# Measures to Increase Immunogenicity of SARS-CoV-2 Vaccines in Solid Organ Transplant Recipients: A Narrative Review

**DOI:** 10.3390/vaccines11121755

**Published:** 2023-11-25

**Authors:** Bo Yu, Christina Tamargo, Daniel C. Brennan, Sam Kant

**Affiliations:** 1Department of Medicine, University of Maryland Medical Center, Midtown Campus, Baltimore, MD 21201, USA; bo.yu@umm.edu; 2Department of Medicine, Johns Hopkins University School of Medicine, Baltimore, MD 21205, USA; ctamarg1@jh.edu; 3Division of Nephrology, Department of Medicine, Johns Hopkins University School of Medicine, Baltimore, MD 21205, USA; dbrenna4@jhmi.edu; 4Comprehensive Transplant Center, Johns Hopkins University School of Medicine, Baltimore, MD 21205, USA

**Keywords:** COVID-19, SARS-CoV-2, vaccine, immunogenicity, organ transplant, kidney transplant

## Abstract

**Purpose of review:** To review the data on the immunogenicity of COVID-19 vaccines, administered by different strategies, in solid organ transplant recipients (SOTRs). **Recent findings**: COVID-19 booster vaccines were given to SOTRs as a widespread practice in many transplant centers, mostly as the third and/or fourth dose in an extended vaccine series, with a significantly improved humoral response compared with the initial two-dose scheme. However, one-third of SOTRs remained unresponsive, despite these boosters. **Next steps**: Vaccination with standard dosing remains the most feasible strategy for attaining protection against COVID-19. Additional booster doses and temporarily holding or reducing mycophenolate mofetil/mycophenolic acid may provide immunogenicity to vaccines, according to recent studies demonstrating some efficacy with these measures. Preexposure prophylaxis with monoclonal antibodies showed benefit in immunocompromised patients but is no longer recommended by the National Institutes of Health (NIH) due to diminished efficacy against Omicron and recent variants. Screening for the presence and titers of SARS-CoV-2-specific antibodies in SOTRs is not recommended in most clinical settings. T cell-based techniques are needed to evaluate vaccine efficacy and risk of infection. As SARS-CoV-2 continues to evolve, new vaccines based on conservative protein component/complexes of the COVID virus, in addition to its spike protein, are warranted to offer prolonged protection.

## 1. Introduction

COVID-19-related morbidity and mortality have declined dramatically since the onset of the pandemic in 2019–2020 and the later Delta variant waves [1]. Many factors have contributed to this, including increased population immunity from widespread vaccination and infection by COVID, the invention of new medical treatments, and the dominance of the Omicron variant and other recent variants, compared to early strains that caused more severe disease [2,3,4,5]. While solid organ transplant recipients (SOTRs) have benefited from these factors, they are at uniquely high risk for COVID-19-related morbidity and mortality, due to immunosuppressive medications and comorbidities [2,6,7,8,9]. The immune responses of SOTRs to commercially available vaccines are often suboptimal because of immunosuppressive therapy, leading to breakthrough infections and subsequent severe illness [10,11]. Most studies thus far have investigated the humoral response as the surrogate marker of vaccine immunogenicity. Patients who received lung transplants due to cystic fibrosis have been reported to have the lowest response, with one study showing that only 23.5% or 52.9%, respectively, developed an IgG (Immunoglobulin G) or IgA (Immunoglobulin A) response after the two mRNA BNT162b2 vaccine doses—the minimum requirement for current mRNA vaccines—compared to a greater than 80% response rate to either dose in patients with cystic fibrosis without lung transplant [12]. In another study of SOTRs, including heart, kidney, liver, and pancreas transplants, slightly more than 30% developed a response after two mRNA vaccines [13]. More recently, one meta-analysis synthesized 44 studies with a total of 6158 SOTRs, the majority of whom received mRNA vaccines, and found that the humoral response rate was 34.2% after the second dose, and up to 65.6% after the third dose, suggesting one-third of SOTRs still had not developed a humoral response with a three-dose vaccination scheme [14].

The cellular immune response (CIR), which rests on the presence of COVID-reactive T cells, may be important for protection against severe disease, given its role in reducing viral load by targeting infected host cells rather than directly recognizing and neutralizing the virus [15,16]. The CIR can also exert protection in the absence of a detectable humoral response [17]. Similar to the humoral response, most studies showed improved cellular response with vaccine boosters, but it is noteworthy that significant heterogeneity exists in the test results and at time points post-vaccination [18].

Despite the less robust response in SOTRs than the general population, evidence has shown that vaccination still provides significant protection for SOTRs and reduced mortality, in the context of evolving COVID variants [19,20,21]. Antibodies produced by COVID-19 vaccination have demonstrated associations with the pneumonia severity and mortality in renal transplant recipients during the Omicron wave; even with lower titers than in the non-immunocompromised patient population, outcomes are better than in the unvaccinated population and are the best in those who achieved an antibody concentration of >100 AU/mL [22,23].

To deliver optimal care to SOTRs amid the constantly changing landscape of COVID-19 variants, it is crucial to utilize strategies for enhancing vaccine immunogenicity within this group. This review summarizes previous research, aiming to help clinicians better utilize current vaccines and other approaches.

## 2. Immunosuppression and Vaccine Response in Solid Organ Transplant Recipients

In addition to vaccines per se, it is important to optimize the factors that are associated with suboptimal immune responses in SOTRs. Physicians may be tempted to modify the immunosuppression regimen or the dosing of immunosuppressants for the enhanced immunogenicity of COVID-19 vaccines. Antimetabolites, calcineurin inhibitors (CNIs), and corticosteroids are the mainstay of immunosuppression for most SOTRs. These medications may have negative influences on the COVID-19 vaccine response, as illustrated in Figure 1.

### 2.1. Antimetabolites

Antimetabolites, especially mycophenolate mofetil (MMF) and mycophenolic acid (MPA), were consistently shown to negatively impact the response to the COVID-19 vaccines in SOTRs in early studies. One meta-analysis showed that antimetabolite use at the time of vaccination, mostly MMF, was associated with a lower response rate, with a pooled odds ratio (OR) of 0.21 [24]. MMF’s negative effect on vaccine response is not limited to SOTRs, as one study showed mycophenolate mofetil was the only independent risk factor associated with seroconversion failure among patients who were taking it for rheumatological diseases [25].

Considering the substantial evidence indicating that MMF can have a negative effect on vaccine responses and indicating the safety of withholding MMF during sepsis, there is a strong inclination to consider suspending the use of MMF in order to enhance vaccine effectiveness. Nonetheless, in a recent randomized study involving kidney transplant recipients (KTRs) who had not seroconverted to their previous two or three mRNA vaccines, discontinuing MMF/MPA (MMF/MPA-) did not result in a higher response rate to mRNA booster vaccinations (third or fourth doses) when compared to those who continued to take MMF/MPA (MMF/MPA+), though MMF/MPA- group showed a subtle tendency toward an increased seroconversion rate by 13% (80% in MMF/MPA- versus 67% in MMF/MPA+, *p* = 0.15) [26]. Additionally, the MMF/MPA- group showed a minor, though statistically insignificant, elevation in antibody titers compared to the MMF/MPA+ group. It is important to highlight that, in this study, the cessation of MMF or MPA occurred one week before and one week after the administration of the booster, and the number of KTRs included in this analysis was lower than the predefined size. No KTRs in the MMF/MPA- group developed acute rejection.

On the other hand, non-randomized studies have shown that reducing or discontinuing antimetabolites in a tacrolimus-based regimen may enhance the effectiveness of COVID vaccines and is considered a safe approach. This is supported by minimal or negligible risks of rejection or the absence of potential markers for rejection (de novo production of HLA antibodies or increased titers of pre-existing donor-specific HLA antibodies, or donor-derived cell-free DNA (dd-cfDNA)) [27,28]. Notably, these studies did not have a control group, so the association between antimetabolite cessation and immunogenicity is hard to establish. Interestingly, a study added detailed insight into the impact of MPA on vaccine response. It revealed a negative linear correlation between the MPA area under the curve (AUC) and the likelihood of seroconversion after receiving two mRNA vaccines. Specifically, for KTRs who were on dual therapy (MPA and prednisone), every 10 mg/h/L increase in the MPA AUC was associated with an adjusted OR for seroconversion of 0.87. For KTRs on triple therapy (CNI + MPA + prednisone) regimens, the adjusted OR for seroconversion was 0.89 [29]. It is worth mentioning that, as of the time of writing this manuscript, there is a US trial still recruiting participants (NCT05060991, Impact of Immunosuppression Adjustment on COVID-19 Vaccination Response in Kidney Transplant Recipients (ADIVKT)). This trial is designed as a prospective, randomized, open-label study, aimed at assessing the impact of reducing or discontinuing MMF/azathioprine on the efficacy of COVID vaccines in KTRs.

In liver transplant recipients, one study found that holding MMF and/or everolimus for two weeks during both the first and second dose of the mRNA-1273 vaccine safely improved the humoral response rate to about 90%, compared to 60.9% in patients on triple therapy with tacrolimus, everolimus, and MMF.

Data for the peri-vaccination suspension or reduction in MMF/MPA in heart and lung transplant recipients are limited. In one preliminary report, heart transplant recipients taking MMF amounts greater than 1000 mg per day had a lower response than those who took less than 1000 mg per day or held MMF [30]. In a preliminary study of stable lung transplant recipients who were over one year post-transplant, the held antimetabolites (mostly MMF, 86% patients) group had a significantly higher anti-spike antibody titer than those who continued with antimetabolites (mostly MMF, 67%) [31].

### 2.2. Calcineurin Inhibitors

CNIs remain the mainstay of immunosuppression regimens for SOTRs. Mixed data exist regarding the effect of tacrolimus on humoral response in SOTRs. In an early study, non-responding liver transplant recipients had higher tacrolimus troughs than responders (6.6 versus 5.4 ng/mL) [32]. Another liver transplant study showed that a high tacrolimus trough (>6.8) was associated with lower antibody titers (42 vs. 309.7 U/L) after mRNA vaccines [33]. A recent study also showed tacrolimus is associated with a poor response in KTRs (OR 0.23, for response in those taking tacrolimus) [34]. However, several meta-analyses showed CNI-based regimens were either not associated or only marginally associated with a low humoral response in SOTRs, compared with non CNI-based regimens [24,35,36,37]. A prospective observational study in Japan also did not show that tacrolimus increased the risk of humoral vaccination failure to the third dose of an mRNA vaccine in KTRs (OR 0.84, CI 0.2–3.52) [38].

A few published studies examined the effect of CNIs on cellular response in SOTRs. A study of liver and heart transplant recipients found no evidence indicating that CNIs or other immunosuppressive medications were associated with a diminished cellular response [39]. A study with a small number of KTRs similarly showed no association between CNI and diminished cellular response [40]. However, a recent meta-analysis showed tacrolimus (vs. a non-tacrolimus regimen) was associated with lower odds of positive cellular immune response (pooled OR 0.5) in KTRs [18].

### 2.3. Mammalian Target of Rapamycin (mTOR) Inhibitors

Mammalian target of rapamycin (mTOR) inhibitors—such as sirolimus and everolimus—were presumed to have less suppression of the vaccine response than CNIs and MPA. One recent prospective study found that mTOR inhibitors (regimens being mTOR + CNI and mTOR + MPA combined) may improve the humoral response in KTRs, with an OR of 7.78 of achieving seroconversion shortly after the second vaccine dose and 7.95 after the third dose (6.40 by multivariate analysis), compared with KTRs on a regimen not including an mTOR inhibitor (CNI + MPA) [41]. However, the evidence level to support the positive effect of mTORs on vaccine response in SOTRs was very weak, and the pooled OR for mTOR to favor a positive humoral response was 1.46 in a meta-analysis [24].

### 2.4. Rituximab, Antithymocyte Globulin, and Belatacept

In this same meta-analysis, rituximab administration within 1 year of COVID vaccination was associated with a negative humoral response (pooled OR 0.21), but it is worth noting that the numbers of KTRs who received rituximab were small, usually less than ten [24]. In a single study involving 43 patients who received rituximab for the purpose of an ABO-incompatible kidney transplant, the OR for a positive humoral response was 0.33 [42]. However, in the revised report, encompassing 131 KTRs, the same Japanese research group found no difference in the rate of humoral response following the second and third doses of an mRNA vaccine, irrespective of whether the patients were on rituximab [43]. Notably, the mean transplant vintage in this study was 5.8 years, and studies with a low response mostly included KTRs who received rituximab within one year prior to vaccines.

Similar detrimental effects on vaccine response were observed in patients who received antithymocyte globulins within one year of vaccination, with a pooled OR of 0.32 [24].

Belatacept was shown to be associated with poor vaccine response in KTRs, compared with other maintenance immunosuppression [44]. In a study of KTRs with belatacept, five percent developed breakthrough infections after three mRNA vaccine doses, and the third dose did not improve the humoral response [45]. Notably, in this study, belatacept infusions were given on the same day as the administration of the vaccine; additionally, all belatacept-treated patients took steroids, and 71% were on MPA as well.

A recent study of kidney transplant recipients receiving belatacept also confirmed the strong negative effect of this medication on the response to the third dose, with an OR of 0.01 for response for those on belatacept compared to those not on belatacept [34]. Another French study showed that regimens including belatacept were associated with a lower rate of humoral response to the third dose (9% with belatacept vs. 40% without belatacept), but that there was no difference after the fourth dose (38% with belatacept vs. 35% without belatacept) [46]. However, more recent studies showed belatacept demonstrated an OR of 0.15 for a positive response after the third dose and of 0.03 after the fourth dose in KTRs [47].

### 2.5. Time from Transplantation and Age at Transplantation

Recently transplanted SOTRs usually require high-intensity immunosuppression, a known risk factor associated with weak responses to vaccinations, including the COVID-19 vaccine [48,49,50]. A German study of KTRs who were transplanted recently (with transplant vintage median of 2 years) and on more intense immunosuppression (77.8% of patients were on a triple therapy of tacrolimus, MMF, and prednisone) had a low seroconversion rate of 34.48% (10/29) after the third mRNA vaccine [51]. Similarly, a cross-sectional study showed that KTRs with a higher transplant vintage had better responses to vaccination than those recently transplanted [52]. Furthermore, a French cohort of heart transplant recipients showed superior responses when vaccinating patients with a long time since their transplant (the average heart transplant age was 17.1 years), on less immunosuppression (with only 46% patients on MMF), and with low CNI trough levels (the mean tacrolimus trough was 5.8 ng/mL) [53].

In a study of pediatric and adolescent KTRs, the third dose of an mRNA vaccine increased seroconversion from 56% after the second dose to 85%; in 16 patients who did not seroconvert with the second dose, 12 (75%) successfully developed antibodies with the third dose [54]. These data were impressive, in contrast to studies in adult KTRs [55]. However, there was no head-to-head comparison between pediatric and adult KTRs, so whether younger SOTRs have better responses to the COVID vaccine than adults is yet to be clarified.

## 3. Vaccine Boosters

### 3.1. Humoral Response to Vaccine Boosters in Solid Organ Transplant Recipients

While most studies published data on mRNA vaccines in SOTRs and evaluated spike S1-specific antibody titers to evaluate vaccine responses, some studies used the specific anti-receptor-binding domain (RBD) antibodies. In these studies, some additionally integrated neutralization assays, to measure the ability of antibodies to counteract various COVID-19 variants [14]. Though not as clinically translatable as outcomes such as infection, hospital admissions, or death due to COVID-19, humoral responses have proven to be good surrogate markers for the immunogenicity or efficacy of vaccines in studies published so far [56].

Early studies of vaccine responsiveness in SOTRs demonstrated limited humoral response to the first one or two doses of the COVID-19 vaccine in kidney, heart, and lung transplant recipients [13,48,57,58]. Vaccine boosters, such as the third or fourth mRNA vaccine doses, can significantly enhance the positivity and titers of COVID-specific antibodies in both the non-immunocompromised population and SOTRs [14,26,59,60,61,62]. Now the CDC recommends the third or fourth mRNA vaccine dose for moderately and severely immunocompromised patients, including SOTRs [63].

Data from KTRs are most abundant among all SOTRs. One study of KTRs showed that a third dose of the BNT162b2 COVID-19 vaccine (tozinameran, Pfizer–BioNTech, New York, NY, USA) increased the overall seropositivity response rate from 37% after the second vaccine dose to 70%, and about half of those who failed to respond to the second dose successfully developed antibodies after the third dose. However, only 27% achieved high titers (defined in this study as 4160 AU/mL), compared to 93% in healthy controls [64]. Another study investigated the third or fourth dose of the mRNA vaccine mRNA-1273 (Moderna, CA, USA) in 36 KTRs, showing a poor seroconversion rate of only 34%, with no KTRs developing neutralizing antibodies to the Omicron variant BA.1 [51].

In a randomized clinical trial, investigators assessed the effectiveness of three distinct booster vaccine strategies in KTRs who did not respond to their initial two mRNA vaccine doses (mRNA-1273 Moderna, or Pfizer, or a combination of both): either a single dose of mRNA-1273, two doses of mRNA-1273, or a single dose of Ad26.COV2.S (Janssen Biotech Inc, Leiden, The Netherlands). This study demonstrated that all three strategies yielded an anti-spike antibody response rate of approximately 68%, with no significant difference observed among these strategies [26]. Also, the concentrations of anti-spike antibodies were comparable across all three approaches and exhibited a significant increase from their baseline levels. In the exploratory analysis, this study found a positive correlation of concentrations of antibodies with neutralizing activity against ancestral, Delta, or Omicron variants, but Omicron variants required higher levels of antibodies for neutralization than the original and Delta strain. On the contrary, an early non-randomized study showed that a heterologous vaccine as the third dose may improve the response rate in KTRs who were previously vaccinated with two mRNA vaccines (42% with the Ad26.COV2.S vaccine, Janssen, Leiden, The Netherlands vs. 35% with an mRNA vaccine, Pfizer-BioNTech, New York, NY, USA) [65].

One study of liver transplant recipients showed that the fourth dose of the BNT162b2mRNA vaccine improved the immune response to the Omicron variant compared to the third dose, as demonstrated by increased RBD IgG and Omicron BA.1 and BA.2 neutralizing antibody levels after the fourth dose. Breakthrough infections occurred in 30.4% of patients following the third dose, in contrast to 18% in those who received the fourth dose, though the difference was not statistically significant [66].

Heart transplant (HTRs) and lung transplant (LTRs) recipients comprise less than 10% of all SOTRs, but their risk of morbidity and mortality from COVID-19 is higher than among KTRs [67]. One study showed antibody positivity increased from 68% to 90% in HTRs, and 43% to 63% in LTRs, one month after the third dose of an mRNA vaccine, but only 59% of HTRs and 25% of LTRs had developed enough antibodies to neutralize the omicron variant [67]. More recent studies with HTRs showed a seroconversion rate increase from 30% to 34% after two doses of a vaccine (BNT162b2, mRNA1273, or AZD 1222), from 57% to 63% after the third vaccine dose [68,69,70], and up to 80.7% with the fourth dose [71].

Lastly, the new bivalent mRNA-1273.214 vaccine (Moderna) has shown superior neutralizing antibody responses against Omicron variants, compared to the widely used prototype mRNA-1273 in phase II and III studies [72]. This updated vaccine may become more and more prevalent and SOTRs are likely to receive it as a booster, but data are limited at this point for SOTRs. A study using the bivalent Omicron BA.4/BA.5 vaccine as the fifth dose in HTRs significantly increased the vaccine’s neutralization ability against the wild type (neutralization titer log2: 86 pre-fifth vaccine vs. 466 post-fifth vaccine), and Omicron BA.1 (14 pre-fifth vaccine vs. 109 post-fifth vaccine), BA.2 (66 pre-fifth vaccine vs. 437 post-fifth vaccine), BA.4 (62 pre-fifth vaccine vs. 319 post-fifth vaccine), and BA.5 (23 pre-fifth vaccine vs. 160 post-fifth vaccine) variants [73]. Despite these data supporting the use of further boosters in non-responders, the yield after the fourth dose needs to be carefully examined. One study reported the response rate longitudinally in KTRs, showing that cumulative humoral response rates in KTRs increased to 19.1% after the second vaccination, 42.0% after the third dose, 74.2% after the fourth, and 88.7% after the fifth dose [47]. Another study showed that the fourth dose only reduced the non-response rate from 24.1% to 18.8%, after the third dose [74].

### 3.2. Cellular Response to the COVID Vaccine in Solid Organ Transplant Recipients

The majority of studies on SOTRs have evaluated the humoral response to COVID vaccine, but only a small proportion evaluated the CIR. In addition, significant inconsistencies exist in testing timepoints after vaccination and in testing methods [39,75,76]. Interferon-gamma releasing assays (IGRAs) or interferon-gamma ELISPOT assays have been used most commonly, though some studies have used intracellular cytokine and/or surface marker staining and detection by flow cytometry [26,77]. Understandably, assessing the CIR poses a greater challenge due to the increased complexity and higher testing expenses compared to antibody tests.

Remarkably, some studies showed that SOTRs have developed cellular immunity and clinical protection even in the absence of humoral responses [78]. This observation suggests that cellular immunity might significantly contribute to protection against severe disease by exerting a cytotoxic effect, thereby reducing the viral load in the host [15,79,80,81]. Another benefit of cellular immunity lies in its ability to potentially counteract mutated viruses through cross-reactivity, giving host prolonged protection against future viral variants. Studies already prove that CIRs, induced by mRNA and adenoviral vaccines, can recognize variants of concern [82]. In contrast, COVID-19 variants often have the ability to evade the humoral immunity generated by vaccines designed for the original strain [83].

Vaccine boosters may improve the CIR in SOTRs as well. In the earlier mentioned randomized trial involving KTRs, after excluding KTRs with a positive ELISPOT result at baseline (after two mRNA vaccines), 42% exhibited de novo positive ELISPOT assay results 28 days after one dose of mRNA-1273, 50% with two doses of mRNA-1273, and 0% with the Ad26.COV2-S vaccine [26]. An observational study of KTRs showed an increased T cell response at 15 weeks after the last vaccine dose (second or third) with a heterologous vector-based vaccine regimen [73]. One recent meta-analysis combining 18 studies showed that the pooled cellular response rate after the second dose of an mRNA vaccine in SOTRs was 48.3%, and 57.6% after the third dose [84]. However, a recent large cohort study showed that a CIR only occurred in 20.4% of infection-naïve KTRs after the fourth dose [74].

In two studies including liver transplant recipients and one of HTRs, the rates of cellular response were increased by repeated vaccines [39,68,75]. However, a liver transplant study showed that only 37% of liver transplant recipients had positive IGRA results 29 days after the second vaccine with mRNA vaccine or AZD1222 [76].

### 3.3. Safety of COVID Vaccine Booster

Most clinical studies of the booster vaccine in SOTRs showed a good safety profile, with most reported side effects being injection site pain, irritation, or other minor side effects. The risk of rejection associated with vaccination is always a concern in SOTRs. Most reports on rejection after the COVID vaccine or COVID infection were case studies, so the actual prevalence of rejection was impossible to evaluate. One meta-analysis evaluated the rejections in SOTRs after the COVID-19 vaccine or COVID-19 infection, showing that 56 organs were rejected post COVID-19 vaccination, and 40 solid organs were rejected after COVID-19 infection. Among the rejections that occurred after vaccination, there were eleven liver cases, six kidney cases, one heart case, and one pancreas case [85]. The risk of rejection associated with COVID-19 vaccination is likely low in relation to the total number of vaccination doses administered and should not discourage physicians from vaccinating SOTRs.

The COVID-19 vaccine may induce de novo DSAs (donor specific antibodies), but this has been limited to case reports and not widely demonstrated. In one cohort of HTRs, with a relatively high seroconversion rate of 94% with three or four doses of BNT162b2 (Pfizer/BioNTech), zero patients developed de novo DSAs after vaccinations [53,86]. In only a few case reports, the COVID-19 vaccine may have induced de novo donor-specific HLA antibodies with the first dose [87].

### 3.4. Role of Vaccine Types

mRNA vaccines have been the most-studied vaccine type in the SOTR population. However, most studies have not compared the efficacy of mRNA vaccines in SOTRs. In one head-to-head study of mRNA COVID-19 vaccines, BNT162b2 seemed to be less effective than mRNA1273 (36% versus 47% response rate, respectively) in SOTRs after the second vaccination [88].

Compared with mRNA vaccines, data regarding other types of COVID-19 vaccines are limited and inconsistent, and there are no direct comparisons published between mRNA vaccines and other vaccine types in SOTRs. One study found that ChAdOx1 (AZD1222, Oxford-AstraZeneca, Oxford, UK), an adenovirus vector vaccine, induced anti-RBD antibodies in only 2.8% and 5.7% of patients after one and two doses, respectively [89]. A study in Thailand showed that KTRs who received two doses of this vaccine had a response rate of 33.33% [90]. A Mexican study of liver transplant recipients showed an antibody positivity of 89.2% with BNT162b2 (Pfizer-BioNTech), 60% with ChAdOx1 nCOV-19 (Oxford-AstraZeneca), 76.9% with CoronaVac (Sinovac, Life Sciences, Beijing, China), 55.6% Ad5-nCov (Cansino, Biologics, Tianjin, China), and 68.2% Gam-COVID-Vac (Sputnik V, Gamaleya National Centre of Epidemiology and Microbiology, Russia), but did not specify the dosing of each vaccine [91]. Another study of HTRs showed an antibody response in 37.5% of patients after two doses of ChAdOx1 and 56% after the third dose with BNT162b2 (Pfizer-BNT) [92].

One study, using the same inactivated whole virion vaccine (BIBP Sinopharm, Beijing, China or CoronaVac, Sinovac, Beijing, China) as the third dose, showed an improved response rate from 5% post-second dose to 43.6% in KTRs. Additionally, 25.6% and 10.3% of patients had developed anti-RBD IgG against the delta and omicron variants, respectively [93]. In a study of KTRs who already had a high seroconversion rate of 92% due to COVID-19 infection prior to vaccination, the inactivated whole virion vaccine CoronaVac (Sinovac, Beijing, China) induced an increase in antibody titers after the first dose, but no further increase with the second dose [94]. One retrospective study examined the efficacy of the inactivated whole virion BIBP vaccine in kidney and liver transplantation and reported a mortality of 0.7%, due to infection caused by delta variant, but this was not a controlled study, so the benefits of the vaccine cannot be evaluated [95].

## 4. Comorbidities Associated with Unfavorable Humoral Response

Besides vaccine and immunosuppression, other patient factors and/or comorbidities have been shown to be associated with a reduced response to the COVID vaccine. For example, in liver transplant recipients, a deceased donor liver transplant, leukopenia, lymphopenia, older age, and chronic kidney disease are associated with vaccine non-responsiveness [33,36,50].

Iron deficiency can cause impaired B cell proliferation and was shown to correlate with low antibody production following the measles vaccine in the general population, leading to the speculation that iron repletion in iron deficient patients may improve the vaccine response [96]. However, one study showed that intravenous iron repletion did not improve the humoral or cellular response to the third COVID vaccine in KTRs, even though it improved iron stores [97]. Diabetes was shown to be an independent risk factor for a suboptimal humoral response to the influenza vaccine, as well as to the COVID-19 vaccine in liver transplant recipients [98,99,100].

## 5. Screening of Vulnerable Patients

A significant amount of SOTRs remain non-responsive to COVID-19 vaccines with boosters [11,26]. Even among responders, antibody titers tend to be low and may be subject to immunity wane and a loss of protection [11,101]. Most transplant centers in the US have access to serological testing for anti-spike protein antibody or anti-RBD antibodies, and Emergency Use Authorization (EUA) guidelines do not explicitly prohibit the use of antibody tests in vaccinated individuals. In certain scenarios, it is appealing to monitor SOTRs for no or low response based on antibody response and titers. Indeed, a study of health care workers in 2021 showed that a total antibody concentration, expressed by binding antibody units (BAU), higher than 1700 BAU/mL provided full protection, whereas a total between 141 and 1700 BAU/mL offered 89.3% protection [102]. In a study of KTRs in the Omicron era, however, an antibody titer higher than 1689 BAU/mL did not lower the risk of hospitalization due to COVID infection, though it did lower the risk for infection (HR 0.41) [103]. The previously mentioned randomized trial showed that the neutralization of the Omicron variant required much higher S1-specific titers [26]. Given that more studies are needed to investigate the correlation of COVID antibodies with protection against infection and severe disease, administrations and societies such as the Infectious Diseases Society of America (IDSA), the US Center for Disease Control and Prevention, and the US Food and Drug Administration actively advise against utilizing COVID-specific antibody tests for screening immunity or making decisions regarding vaccination [104,105,106].

One study recommended monitoring for CIRs in immunosuppressed patients [107]. However, several technical difficulties are expected before CIR monitoring can be applied to determine protective immunity. First and foremost, the CIR exerts its effects via cytotoxic T lymphocyte recognition and the attack of virus-infected cells and facilitation of humoral immunity. Consequently, the contribution and correlation of the CIR to clinical protection are hard to establish [82]. Second, CIRs are highly heterogeneous in specificity and quantity among vaccine recipients, and the effects of immunosuppression can further complicate the SOTRs’ CIR to vaccines. Lastly, the complexity and cost of the measurement of the T cell response precludes a routine clinical implementation. As of now, there are no high-quality trials accessible to ascertain the relationship between CIR and clinical protection.

## 6. Pre-Exposure Prophylaxis

Monoclonal antibodies have been developed for pre- and post-exposure prophylaxis, as well as for the treatment of COVID. Most published studies utilized tixagevimab–cilgavimab, an extended half-life monoclonal antibody combination derived from cells obtained from a patient infected with a COVID strain [108]. Tixagevimab–cilgavimab can attach to numerous epitopes within the receptor-binding domain of the COVID spike protein, effectively neutralizing the virus [109]. The FDA approved its use in high-risk populations via EUA [110]. As a prophylactic measure, data supporting the use of pre-exposure prophylaxis (PrEP) are less robust than for vaccines, with only one randomized controlled trial available thus far [111]. This study, PROVENT, recruited a total of 5973 participants, with the last injection administered on 29 March 2021, and showed a relative risk (RR) reduction of 76.7% for the first symptomatic COVID infection, compared with the administration of a placebo after a median of 83 days from tixagevimab–cilgavimab (150 mg/150 mg) administration. No COVID-related deaths occurred in the treatment group, but the placebo group experienced two deaths due to COVID [111]. This dosage recommendation was originally established when tixagevimab–cilgavimab received its initial approval, primarily targeting the ancestral strain of COVID. Notably, on February 24, 2022, the FDA updated its suggestion to increase the dose from 150 mg of tixagevimab and 150 mg of cilgavimab to 300 mg of each. These adjustments were made due to the reduced neutralizing effectiveness of the medication against the emerging Omicron variants (BA.1–BA.5) [112].

Data regarding the safety and efficacy of tixagevimab and cilgavimab in SOTRs are scarce. One study demonstrated that tixagevimab–cilgavimab PrEP was associated with a lower breakthrough infection rate in SOTRs who received at least one dose of a COVID vaccine in the early Omicron wave than in the control group (5% in treatment vs. 14% in control group) [113]. Another study used self-reported outcomes by SOTRs who received three or more COVID vaccines and showed that tixagevimab–cilgavimab PrEP had good tolerability and a good safety profile, with a breakthrough rate of 8.9% in 3 months of follow-up, mostly within the time period of BA.2 and BA.4/5 sublineage predominance [112]. This study did not have a control group, but patients who received a lower dose (150 mg/150 mg) appeared to have more breakthrough infections than those who received a higher dose (300 mg/300 mg) (20/189 vs. 16/203, respectively) [112]. A real-life study of KTRs during the Omicron wave with BA.1 and BA.2 dominance showed that 12.3% of KTRs who received 150 mg/150 mg tixagevimab–cilgavimab developed symptomatic COVID. This is in comparison to 43.3% of KTRs who did not receive tixagevimab–cilgavimab, indicating that a higher dosage may be necessary [114]. Another KTR study found that tixagevimab–cilgavimab, at a 300 mg/300 mg dose, in individuals with low antibody levels (less than 264 BAU/mL after receiving three mRNA vaccines), offered comparable protection against symptomatic Omicron infection, hospitalization, or ICU admission, as well as mortality, when compared to the level of protection achieved in individuals with “protective antibody levels” (greater than 264 BAU/mL) induced by three mRNA vaccine doses [115]. In a study involving various immunocompromised patients, it was found that individuals receiving tixagevimab–cilgavimab at a 300 mg/300 mg dose had a 92% reduced risk of hospitalization or mortality, compared to those not on PrEP. However, the study did not provide information regarding the number of transplant patients included in the analysis [116].

Conversely, two recent studies with LTRs during the Omicron wave demonstrated that tixagevimab–cilgavimab (with most patients receiving a 300 mg/300 mg dose) did not significantly alter the rates of hospitalization, severe disease, or mortality (ranging from 1% to 11.8%) when compared to a placebo, even though tixagevimab–cilgavimab did reduce the incidence of symptomatic COVID infection in both studies [117,118]. Of note, 70–89% of these LTRs were fully vaccinated with at least two doses in these studies.

Monoclonal antibodies as a measure for COVID PrEP face many challenges. First, the prohibitive cost, limited availability of the medication, and infusion facilities restrict their widespread and prompt administration. Second, the safety profile of monoclonal antibodies is not established as well as vaccines or small molecule antivirals such as nirmatrelvir. However, the most difficult challenge may be the decreasing efficacy of PrEP to combat ever-evolving variants. In vitro studies have shown the reduced neutralization ability of tixagevimab–cilgavimab to Omicron BA.1 and BA.2, and even more reduced neutralization to recent variants such as BQ 1.1 and XBB 1.5 [119]. Another in vitro study showed that, while tixagevimab–cilgavimab continued to inhibit BA.2.12.1, BA.4, and BA.5, the titers needed to achieve an equivalent level of inhibition (50% neutralization) for BA.5 were approximately 30.7 times higher than those required for the ancestral strain [120]. Nevertheless, the previously mentioned clinical data continued to demonstrate the effectiveness of tixagevimab–cilgavimab during the Omicron wave [112,113,114,115]. Efficient and accurate tests that can predict the in vivo efficacy of PrEP are urgently needed in the setting of ever-evolving variants. The most recent revisions in the FDA Fact Sheet for tixagevimab–cilgavimab indicate that variants bearing spike substitutions like R346T or K444T, in conjunction with F486S or F586V, exhibit resistance to neutralization. These substitutions are found in variants such as BA.5.26, BF.7, BF.11, BJ.1, BN.1, and XBB [121].

The current literature review has shown conflicting data regarding the efficacy of PrEP in SOTRs. Given the diminishing efficacy of available PrEP in the context of emerging variants, mostly subvariants of Omicron, the NIH recommended against the use of tixagevimab–cilgavimab as PrEP for COVID, and the FDA no longer authorizes its use for this purpose [122].

Modified monoclonal antibody medications are currently being assessed in the SUPERNOVA trial (NCT0564810), with active patient enrollment at the time of writing this review. Pharmaceutical companies are also working on production platforms that can promptly modify the target antigen, allowing the medication to keep up with the evolution of variants.

## 7. Immunization of Close Contacts and Patients on Transplant Wait Lists

Patients with chronic liver disease or end-stage kidney disease on hemodialysis or peritoneal dialysis showed impaired immune responses compared to healthy controls [123], but invariably had better response rates than SOTRs, leading many transplant centers to mandate vaccination for their patients on the waiting list [124,125,126,127,128]. Whereas vaccination prior to being exposed to intense immunosuppression can theoretically maximize the protection offered by vaccines, a mandate to vaccinate while on the wait list entails complicated ethical considerations and is, thus, a subject of in-depth discussion far beyond the scope of this review [129,130].

COVID transmission in the household setting is an important issue for SOTRs given the large number of household cases and the higher secondary attack risk by the Omicron variant compared to the Delta variant [131]. Consequently, COVID-19 vaccination of patients’ household members and health care staff involved in patient care, namely ring vaccination, might be desirable, as healthy people around transplant recipients usually have good vaccination responses and may help to build up barriers to prevent transmission [132,133].

## 8. Limitations of this Review and Next Steps

In general, the evidence level of the literature on vaccination in SOTRs in this review is low, due to the non-randomized and observational designs. This is due to the technical challenge that arises from the much lower number of SOTRs compared with the healthy population. While only a limited number of studies were randomized, it is important to note that the sample size of study subjects might be too underpowered to detect differences in vaccine immunogenicity caused by various interventions, such as additional booster doses or adjustments in immunosuppression.

As we enter the fourth year since the COVID-19 outbreak in late 2019, the virus has undergone substantial mutations. These mutations have led to variants that are either not susceptible or only minimally susceptible to antibodies produced by the vaccine designed for the original strain, and monoclonal antibody PrEP. This may compromise the validity of data previously reported. Innovative vaccine technologies, particularly those utilizing peptides or epitopes designed to stimulate T cell responses, have the potential to protect patients from new variants. This is due to the T cell response’s longevity and its broader reactivity to the relatively conservative epitopes/protein complexes of various variants [134].

Most studies assessing COVID vaccine responses in SOTRs have disparate protocols, so heterogeneity is obvious in patient demographics and the evaluation process. Delayed positivity may partly explain the variation in vaccine responses in SOTRs due to different testing times post-vaccination, as data had shown that seroconversion in SOTRs occurred as late as six months after the administration of the COVID-19 vaccine [135,136]. It would be desirable for future studies to use uniform time points to test SOTRs for responses. Moreover, in future trials, real-world endpoints, like the incidence of severe disease and hospitalizations, may prove to be more practical measures than antibody titers for assessing vaccine immunogenicity.

It is reasonable to contemplate temporarily halting the use of antimetabolites as a measure to boost the vaccine response. Nevertheless, the antimetabolite suspension protocol still requires optimization, as a brief suspension may not be adequate for achieving the desired immune response. Conversely, an extended suspension may elevate the risk of rejection and potentially harm long-term graft function and outcomes. Thus, peri-vaccination suspension of MMF/MPA should not be practiced as a routine measure until more data are available.

The anti-spike antibody test should not be used routinely in clinical settings to confirm protection. Nevertheless, it could be a practical approach for identifying vaccine non-responders among SOTRs and implementing a multi-modal preventive strategy, which may include updated PrEP or enhanced patient protection measures.

Monitoring individuals at elevated risk, such as those who have recently undergone transplants, to detect refractory non-responsiveness to vaccines could enable the implementation of preventive measures, adjustments to the vaccine scheme, and modifications to immunosuppressive regimens. CIR monitoring may hold more clinical relevance due to its broader reactivity than humoral response and longevity of memory T cells, making it a more promising correlate of protection against future variants. However, the methodology to monitor CIR still needs standardization and improvement before it can be widely used in clinical practice.

## 9. Conclusions

Strategies to improve immunogenicity are highly warranted in SOTRs. Holding or reducing MMF or MPA for a short period of time peri-vaccination did not improve mRNA COVID vaccine responsiveness in KTRs in a randomized controlled trial, though less robust data showed safety and efficacy in all types of SOTRs. The American Society of Transplantation, American Society of Transplant Surgeons, and International Society of Heart and Lung Transplant issued a joint statement in August 2021 that indicated there is no reliable guide to modify a immunosuppression regimen to prepare for a vaccine response [136].

There is a need for novel vaccine technology that can offer extensive protection resistant to frequent viral mutations. The NIH and FDA currently recommend against the use of tixagevimab–cilgavimab as COVID PrEP, given its diminished efficacy against new variants. Currently, the most efficient method for protecting SOTRs against COVID-19 is to enhance the immunogenicity of existing vaccines with standard booster doses (three or four doses). We anticipate that the ongoing randomized trial will provide additional data to enhance our understanding of the safety and efficacy of modifying immunosuppression to improve vaccine response.

## Figures and Tables

**Figure 1 vaccines-11-01755-f001:**
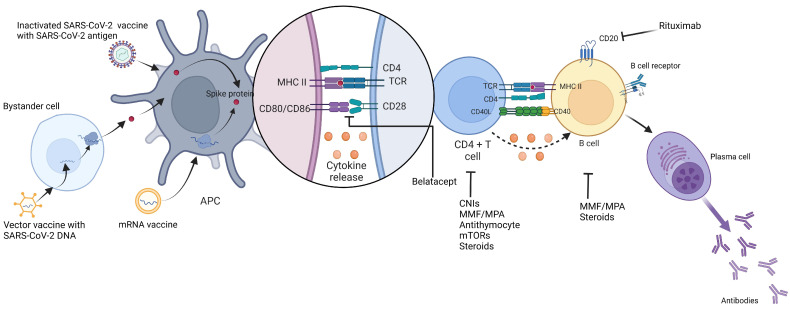
COVID-19(SARS-CoV-2) vaccine, generation of immunogenicity, and effects of immunosuppression on vaccine response. APC: antigen presenting cells, MHC: major histocompatibility complex, CNIs: calcineurin inhibitors, MMF: mycophenolate, MPA: mycophenolic acid, mTORs: mammalian target of rapamycin, TCR: T cell receptor.

## Data Availability

Not applicable.

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
