# Peer review of "Measures to Increase Immunogenicity of SARS-CoV-2 Vaccines in Solid Organ Transplant Recipients: A Narrative Review"

_vaccines, 2023, doi:10.3390/vaccines11121755_

Round 1

Reviewer 1 Report (Previous Reviewer 1)

Comments and Suggestions for Authors

Thank you for revising the paper.  The paper reads much better and is a more accurate reflection of the existing data.  One point that still needs further clarification.

The authors state "Infectious Disease Society of America(IDSA), the US Center for Disease Control, and the US Food and Drug Administration do not endorse the routine practice of conducting COVID-specific antibody tests" after vaccine.  The FDA actively discourages this practice which is not well captured in this statement.  

Author Response

Thank you for revising the paper.  The paper reads much better and is a more accurate reflection of the existing data.  One point that still needs further clarification.

The authors state "Infectious Disease Society of America(IDSA), the US Center for Disease Control, and the US Food and Drug Administration do not endorse the routine practice of conducting COVID-specific antibody tests" after vaccine.  The FDA actively discourages this practice which is not well captured in this statement. 

Clarified, line 404-409. Thank you!

Reviewer 2 Report (Previous Reviewer 2)

Comments and Suggestions for Authors

Yu et al. reviewed the data on the immunogenicity of COVID-19 vaccines administered by different strategies in solid organ transplant recipients (SOTRs). The review is well-organized and written very well. However, there are several issues that should be addressed:

  1. 1. All the references should be revised carefully, and the dot should be after the reference number, not before. For example, [1] not .[1].

  2. 2. There are several typos and spaces that should be revised. For example, lines 21, 32, 76.

  3. 3. Immunosuppression and vaccine response in SOTRs should be presented by a figure to be more easily understandable for the readers.

  4. 4. The limitations of the study should be provided.

  5. 5. The entire review should be revised very carefully, as it contains many grammar mistakes.

Comments on the Quality of English Language

 Moderate editing of English language required

Author Response

  1. All the references should be revised carefully, and the dot should be after the reference number, not before. For example, [1] not .[1].

Fixed. Thank you for pointing out this problem.

  1. There are several typos and spaces that should be revised. For example, lines 21, 32, 76.

Fixed. Thank you!

  1. Immunosuppression and vaccine response in SOTRs should be presented by a figure to be more easily understandable for the readers.

Addressed: Line 91-96, Figure 1. Thanks for this invaluable suggestion!

  1. The limitations of the study should be provided.

Line 503, added limitations of this review and combined them with next steps.

  1. The entire review should be revised very carefully, as it contains many grammar mistakes.

Double checked grammar and typos, thank you!

This manuscript is a resubmission of an earlier submission. The following is a list of the peer review reports and author responses from that submission.

Round 1

Reviewer 1 Report

Comments and Suggestions for Authors

Thank you for allowing me to review your paper entitled "Measures to Increase Immunogenicity of SARS-CoV-2 Vaccines in Solid Organ Transplant Recipients: A Narrative Review."  This paper attempts to review the data to improve vaccine responses in SOT recipients.  The paper lacks discussion of some key topics, over emphasizes others and overall presents a suboptimal review of the literature.  

Introduction:  The discussion of poor response after the first dose is misleading as for all mRNA vaccines 2 doses at a minimum are required.  Would remove.  Introduction includes no mention of cellular response and their contribution to protection vs. severe disease and death and that a much higher rate that serologic immunity.

Immunosuppression:  There is a long discussion around MMF but minimal discussion about other agents - would expand on the - esp on the ritixumab and belatacept - there is a large body of data that is only superficially covered in the review.

The discussion about reduction of MPA is highly skewed.  There is a randomized trial that failed to show a benefit - spending tons of time on small often uncontrolled studies and only mentioning the data from the randomized trial is misleading.  Would comment on ongoing US study.

Booster doses - there is again extensive discussion of boosters but limited discussion about reducing yield after the 3rd dose.   Limited data of boosting of cellular responses in this section even when data from randomized trial data are available.

Screening of response:  Nearly all transplant centers have access to serologic testing so the statement is not accurate.  No discussion about screening for cellular response.  No statement that checking serology is not recommended by professional societies and FDA.  

PREP was almost superfluous and not complete.  Would discus limited data and challenges as well as potential benefit.  May even want to focus on who would benefit from MAbs.   

Reviewer 2 Report

Comments and Suggestions for Authors

Yu et al demonstrated the immunogenicity of COVID-19 vaccines, administered by different strategies, in solid organ transplant recipients. The review is very interesting. However, there are some concerns that should be addressed.

  1. The reference style should be modified according to the Journal instructions.
  2. Authors should provide the Title "Types of COVID-19 vaccines" in order to enrich the review.
  3. Antimetabolites, calcineurin inhibitors, and corticosteroids are the mainstay of immunosuppression for most SOTRs. Authors should provide schematic figures for this immunosuppression mechanisms and how they are working.
  4. The abbreviation should be carefully revised and a list of abbreviations should be provided.
  5. There are many typo-errors; the review should be revised carefully. For example line 23.37, 130, 134, 271
  6. There are many grammar mistakes, the review should be revised carefully.
Comments on the Quality of English Language

Reviewer 3 Report

Comments and Suggestions for Authors

The submitted review is comprehensive, well structured and summarizes important data concerning SARS-CoV-2 specific vaccine responses in SOT patients.

Comments on the Quality of English Language

Use of English language is overall proper and mostly clear, however the manuscript requires thorough check of grammar, in particular use of  pronouns.

Reviewer 4 Report

Comments and Suggestions for Authors

This review describes immunogenicity of SARS-CaV-2 vaccination in solid organ transplant recipients. It is well written and cover widely including both humoral and T-cell response, various organ transplantations, many types of vaccine, and virus variants. Immunogenicity showed less responsive in many organ transplantations and one of the reasons is antimetabolite immunosuppression. The review indicates temporary holding and reducing antimetabolites improves immunogenicity in organ transplant recipients. This is important message. The review also indicated the importance of measuring neutralizing antibody titer in organ transplant recipients. I agree with this opinion. 

Minor comments

1.     Please take “]” off in P.5 L.228.

2.     Period disappears before “In” in P.6 L.272. 

3.     Missing words after “in” in P.6 L.288.